# Selection of Method of Chemical Analysis in Measuring the Salinity of Mineral Materials

**DOI:** 10.3390/ma13030559

**Published:** 2020-01-24

**Authors:** Teresa Stryszewska, Marta Dudek

**Affiliations:** Faculty of Civil Engineering, Cracow University of Technology, Cracow, Poland; marta.dudek@pk.edu.pl

**Keywords:** salinity of materials, test methods, historical buildings, methods UV, HPLC, EDS, XRF

## Abstract

The article deals with the issue of salt content in brick buildings, which plays an important role in the assessment of the technical condition, in particular of historic buildings. A question has been asked about the selection of the best research method to determine the salinity of mineral materials. To obtain the answer, the authors conducted some tests on ceramic bricks salted with seven types of salt solutions. Research methods such as: spectrophotometry (UV), ion chromatography (High Performance Liquid Chromatography (HPLC)), X-Ray Fluorescence Spectrometry (XRF) and Energy-Dispersive X-Ray Spectroscopy (EDS) were compared. The above methods belong to two groups: the first is based on aqueous extracts and allows the determination of water-soluble salts, and the second concerns testing directly on the sample so that insoluble salts can also be determined. The results tests indicate that the methods based on solid phases (XRF and EDS) give higher salinity values than those based on aqueous extracts (HPLC and UV). The results were also analysed with regard to the type of salt. Larger differences are observed for sulphate salts while chloride salts are characterised by smaller differences. On this basis, it is concluded that the salt content of the material is best assessed using tests that make it possible to recognise the salt in question and its quantity.

## 1. Introduction

Determination of the level of salinity of buildings is an important element in the assessment of the technical condition of engineering structures. It allows the identification and measurement of the progress of the corrosion processes in the material [1,2,3]. Determination of the content of harmful substances is a necessary stage preceding renovation and conservation works in historic buildings [1,4]. There is no clear indication of the method that best reflects the actual salinity of the building materials in use. The commonly applied test methods can be divided into two groups depending on the way in which the sample is prepared. The first of them comprises tests for the content of substances and salts that are soluble in water. The second group are methods that allow determining the content of all (soluble and insoluble) components. The first group comprises methods based on the examination of aqueous extracts. Therefore, for tests of this type the sample has to be ground, mixed with distilled water and left for the soluble components to be leached out. The result obtained by these methods specifies the content of salts and water-soluble substances passed into the solution [4,5]. In this case, the result is usually expressed as the ion concentration per unit volume of the solution. This group includes titration, colorimetry, conductometry, spectrophotometry and ion chromatography. The other group of methods includes tests using samples in the solid state. Preparation of the sample typically consists of powdering it. In this group of methods, all types of salts and substances, both soluble and insoluble in water, are recorded. The result is usually a quantitative elemental composition expressed as percentage by weight. The most popular methods in this group are X-ray fluorescence (XRF) and elemental composition analysis by scanning electron microscopy (SEM) with energy-dispersive X-ray spectrometry (EDS), together with an image analysis. 

The choice of the test method implies the nature of the result and is closely related to its value. When choosing the test method, it is also necessary to take into account the precision, accuracy if the determinations and the measurement range, the sample size (including the possibility of preparing a representative sample), and the time and cost of analysis. 

According WTA guidelines [1,6,7], the recommended method for determining the salt content in walls is by testing aqueous extracts in which SO_4_^2−^, Cl^−^ and NO_3_^−^ ion contents are determined. This approach simplifies the issue of the aggressiveness of salt to the ceramic material. Based solely on the results of tests for the content of salt anions in extracts, we are not able to reliably assess the threat to the durability of the material as identification of the type of salt is essentially important [8,9]. Moreover, salt solubility is not a measure of aggressiveness of a given solution although it is true that salts with the highest solubility are more dangerous [10]. The main and direct cause of salt aggressiveness towards the mineral material is not the salt solution itself but first of all crystallisation of salt from this solution [8,11,12,13,14,15].

A wider approach to the assessment of salinity of mineral materials is proposed in “A guide to salt damp in historic and older buildings” [2], according to which the determination of salt content in materials should be based on a full chemical analysis of cations and anions using, e.g., ion chromatography for the cations and coupled plasma atomic emission spectrometry for the anions. Additionally, full identification of the type and amount of salt facilitates the selection of methods for desalination of walls, which in consequence allows for an effective process of their drying and carrying out conservation work [9,11,16,17,18,19]. Depending on the type and content of salt, the process of degradation of functional properties takes place with different intensity [8].

The purpose of this paper is to assess the usefulness of testing and analytical methods in examining the salinity of mineral building materials and to try to indicate one of them as the most reliable. Studies were carried out on bricks exposed to chloride, sulphate and nitrate salts in laboratory conditions. The salinity of the bricks was determined by spectrophotometry, high-performance liquid chromatography, X-ray fluorescence and X-ray energy dispersion analysis in a scanning microscope together with microstructure observation. The test techniques were selected in such manner as to obtain a wide range of results with regard to both the content of soluble and insoluble substances and phase identification.

## 2. Methods Used in Salinity Test of Building Materials and Test Apparatus

### 2.1. Methods Using Aqueous Extracts

#### 2.1.1. UV Spectrophotometry

Spectrophotometry is a quantitative measurement of light transmission or reflection by a sample in the form of a solution. A light beam passes through the sample in the tray and falls on the detector. Then, the spectrum of the sample being tested can be recorded. Spectrophotometric determination of the concentration of the substance in solution is based on the absorbance of the solution according to the Lambert–Beer law, which determines the relation between the permeability and absorption of light propagating through the solution being tested. Test result using this method is a qualitative (presence) and quantitative (concentration) analysis of the substance in solution. The concentration is the basic factor determining the amount of light absorbed at a given wavelength for a particular substance.

The spectrophotometric tests presented in the paper were performed with the use of a UV–Vis V-630 spectrophotometer (JASCO, Tokyo, Japan). Preparation of the sample consisted of fragmenting it (by grinding in a ball mill for grains below 0.08 mm) and mixing it with distilled water in a mass ratio of 1:10 (10 g of ground sample and 100 g of water). For water-soluble substances to be leached out from the test sample, the mixture was left for 48 h, stirring it every 12 h (10–15 min). After that time, the mixture was filtered through a filter paper and analysed in a spectrophotometer. SO_4_^2−^ ion was determined at (λ) wavelength of 450 nm, chloride at (λ) wavelength of 455 nm and nitrate at (λ) wavelength of 410 nm. The result obtained was expressed in mg/dm^3^. The extracts prepared in this way were also used in the liquid chromatography test. 

#### 2.1.2. Ion Chromatography

Ion chromatography is a method of separating mixtures into individual components to identify and quantify them. The basis for the separation are the differences in the behaviour of the compounds in a two-phase system in which one is stationary and the other moves relative to the first one. For liquid chromatography (HPLC), the mobile phase is a liquid. It is a method which allows determining, quantitatively and qualitatively, the content of substances even at their low concentration and in the presence of other compounds [20]. The direct result is a chromatogram, the analysis of which allows assessment of the composition of the sample. 

The determinations were performed using the Dionex chromatographic system (ICS-2000, Dionex Corporation, California, USA) consisting of two ICS-2000 (Dionex Corporation, California, USA) chromatographs operating simultaneously and working with an AS-40 auto-sampler. The chromatograph responsible for the anion analysis (anion module) was equipped with AG-18 and AS-18 4mm columns and an AERS 500 4 mm suppressor with a KOH gradient elution. The cationic module was equipped with CG-16 and CS-16 5mm columns and a CERS 500 4mm suppressor with MSA isocratic elution.

### 2.2. Methods Using Powder Preparations

#### 2.2.1. X-Ray Fluorescence

X-ray fluorescence (XRF) is a sensitive analytical method for determining elemental composition of a sample. The examination involves excitation of characteristic X-ray radiation by means of radiation from an X-ray tube or a synchrotron and measurement of the resulting X-ray spectra. The basis for the qualitative and quantitative analysis is the element-specific X-ray radiation. The high sensitivity of this method makes it possible to detect even traces (at the ppm level) of light elements such as boron, carbon, nitrogen, etc. The direct result is a fluorescent spectrum. Preparation of the sample for the test consists in grinding the material, and then compacting it into a tablet under high pressure. 

The examination of the elemental composition of the materials tested using X-ray fluorescence was performed on PANalytical WDXRF Axios mAX spectrometer (Malvern Panalytical, Malvern, United Kingdom) equipped with a 4 kW Rh lamp. The test was performed on samples prepared in the form of pellets (in diameter 32 mm). The used pressure for pressing was 15 MPa. The identification and quantitative analysis of the elemental composition was based on the measurement of the wavelength (of characteristic radiation) emitted from the sample. The surface area of the test sample was 3 cm^2^.

#### 2.2.2. Energy-Dispersive X-ray Spectroscopy

X-ray microanalysis enables qualitative and quantitative analysis of the elemental composition of a sample. The principle of determination of the elemental composition by the EDS method is based on the penetration of the sample with a beam of primary electrons emitted from an electron gun. These electrons hit the surface of the sample and excite it. As a result, the material emits characteristic X-rays, i.e. quanta of energy characteristic for a particular element. These signals are recorded with an EDS detector and processed into the qualitative and quantitative composition. The direct result is the spectrum of energy bands of the individual elements included in the composition of the test material. It is then processed into the qualitative and quantitative composition of the sample. One of the conditions for a reliable analysis is an appropriate preparation of the surface of the sample which must be perfectly smooth (it is usually polished). Furthermore, in view of the limited test area, it seems appropriate to grind and prepare the sample in the form of compressed powder when determining the composition of heterogeneous samples. With such preparation, the result will be representative for the material tested [21].

The elemental composition analysis by the EDS method presented in this paper was performed using Carl Zeiss EVO MA10 scanning electron microscope (SEM) (Carl Zeiss Microscopy, Jena, Germany) with Bruker EDS XFLASH 6/30 detector (Bruker, Hamburg, Germany) recording characteristic X-ray radiation, at an acceleration voltage of 20 kV and a work distance (WD) of approx. 10 mm. The test was carried out in a variable vacuum in the range of 80 to 120 Pa on non-sputtered samples. Quantitative analysis of elemental composition was performed based on multipoint scanning of the sample surface. The samples tested were those used in the X-ray fluorescence test.

## 3. Test Materials and Method of Saturating Samples with Salt

Due to the high inertness and the lowest possible reactivity to the external environment, ceramic bricks were chosen for the testing. Such a matrix limits the reactions between its components and salts. The bricks were dried to constant weight at 105 °C, and then samples in the shape of cylinders with a diameter of 5 cm and the same height were cut out. In total, 40 cylinders were cut out for the whole test (5 for each salt and reference material). The cut out cylinders were stored for 2 weeks in a laboratory room at 22° ± 1° and relative humidity (RH) 35% ± 5% until the air-dry condition was achieved. The samples were then salted by cyclic saturation with salt solutions. Each cycle comprised 2 d of saturation by capillary action (after that time no further weight gain was recorded) and 8 d of drying under laboratory conditions at 22° ± 1° and relative humidity (RH) 35% ± 5%. The samples were saturated by soaking in 10% solutions of MgCl_2_, KCl, NaCl, MgSO_4_, K_2_SO_4_, and Na_2_SO_4_, and Mg(NO_3_)_2_. A total of five cycles of 10 d each were performed. In addition, 1 cycle of saturation with distilled water was performed. Figure 1 shows the layout of the experiment (saturation of the material with salt) and the tests performed after the end of the salt saturation of the ceramic samples. During the experiment (salt saturation) the samples were weighed every 24 h. 

## 4. Results of Mass Changes of Ceramic Samples after Cyclic Saturation with Salt Solutions

The results of the investigation into the effect of the presence of chloride and sulphate salts (Na, K and Mg) and magnesium nitrate on the drying process of bricks are shown in Figure 2. The graph shows the changes in the mass of ceramic brick samples during the experiment; increase in the mass of samples during soaking (phase I) and loss of mass during drying (phase II) for each salt. The graph also shows the reference curve determined for bricks soaked in distilled water.

The process of drying ceramic bricks saturated with distilled water under the set laboratory conditions leads to complete water removal. The presence of salt, on the other hand, clearly limits the possibility of this material drying out. Sodium sulphate has the least influence on the drying process. At the end of the experiment, i.e., after five cycles, the weight increase for the sample saturated with this salt is 3.3%. On the other hand, the greatest changes were recorded for potassium sulphate; weight gain of the sample saturated with this salt after the end of the experiment is 13.3%. A large weight gain was also recorded for magnesium sulphate, which is 11%. The remaining salts, including all magnesium chloride and nitrate salts, have a weight gain of 7.5–9.5%.

## 5. Results of Chemical Analysis Tests

The spectrophotometric examination (UV) and liquid ion exchange chromatography (HPLC) were performed on aqueous extracts. For this purpose, after five soaking and drying cycles the samples were ground. Then, distilled water was added at a mass ratio of 1:10 and left for 48 h (stirring every 12 h). After that time, the mixtures were filtered and aqueous extracts were obtained for testing. The spectrophotometric examination determined the content of sulphate ions SO_4_^2−^, chloride ions Cl^−^ and nitrate ions NO_3_^−^. In the ion chromatography examination, a wide spectrum of ions was determined; however, the paper only presents the results for the content of those ions that are the basis for determining the level of salinity of mineral building materials. In both methods the results obtained were expressed in [mg/dm^3^]; these were then converted relative to the sample mass and expressed as [%] of the mass. 

The X-ray fluorescence (XRF) examination and the energy-dispersive X-ray spectrometry (EDS) were performed on powdered samples (the samples were ground in a ball mill) prepared in the form of tablets (powder pressed in a hydraulic press). In both methods, the result is the elemental composition of the materials tested, expressed as [%] of the sample weight. The sulphur content has been converted to SO_4_^2−^ ion. The paper only presents the results pertaining to the harmful substances, i.e. chlorides and sulphates. 

The test results presented in the paper (Table 1 and Figure 3) are the averaged results of three replicate measurements.

Due to the clear variability of the results obtained in the examination of the extracts and the powder samples, the content of sulphates and chlorides in the sediments left by the aqueous extracts was tested. For this purpose, the sediments were dried to a constant mass at 25 °C. Then, the samples were analysed by XRF and EDS. The results obtained together with those in Table 1 are shown in Figure 2. 

The results obtained showed that the sum of sulphates and chlorides determined in aqueous extracts and in sediments left by the extracts is the total content of these ions equal to the S and Cl content that was determined in powder samples by XRF and EDS.

## 6. Results of SEM Observations

The surfaces of the samples on which salts crystallised were observed using SEM. The results of the observations of the microstructure of the crystallised salts together with an EDS analysis of the elemental composition are presented in Figure 4, Figure 5, Figure 6, Figure 7, Figure 8, Figure 9 and Figure 10. Based on the EDS analysis, it was found that the salts which crystallise in the ceramic matrix are mainly the same ones that were used to saturate the bricks. The exception is a brick exposed to magnesium chloride solution in which KCl crystals were observed in addition to MgCl_2_ crystals.

The basic difference in the microstructure of the salts formed is the size of the crystals and their shape. The finest are sodium sulphate crystals. They take the form of oval fine crystals with dimensions not exceeding a few μm. Magnesium sulphate crystals are also fine. Potassium sulphate crystals, which is well crystallised with clear edges, has an interesting form. Their size ranges from several micrometres to ~150 μm. The largest are magnesium nitrate crystals, which can be even 2–3 mm long. They crystallise in two forms; the first one (the dominant one) is characterised by elongated, oval shapes and the second one by spherical forms with a diameter of up to ~100 μm (Figure 10).

## 7. Discussion of Results

As stated in the introduction, the test methods applied in the paper can be divided into two main groups. One of them comprises methods for testing aqueous extracts, and the other one comprises methods for solid samples (powdered samples). The results obtained clearly indicate differences in sulphate and chloride contents depending on the test method applied. In all cases, the results obtained by X-ray fluorescence (XRF) and energy-dispersive X-ray spectroscopy (EDS) for solid-phase samples are higher than those obtained by ion chromatography (HPLC) and spectrophotometry (UV) for water extracts. For chloride salts, the differences are much smaller, while for sulphate salts, the differences are significantly higher. Note that the results of tests of salt content in liquids made by the two methods are very similar, and likewise the results of tests of solid samples are also very similar. For the NaCl and KCl salts, the results for chloride content determined by the UV and HPLC are almost identical, and so are the results of the chloride content obtained by EDS and XRF. For example, the chloride ion contents in the sample exposed to NaCl salt determined in the aqueous extract are 1.29% and 1.38% of the mass while for the powdered sample they are 2.43% and 2.49% of the mass. The results of tests on samples exposed to sulphate salts are also similar; the results of UV and HPPLC tests for aqueous extracts are very similar like the results of solid-phase tests with EDS and XRF, which are similar too. Nevertheless, the differences between the results depending on the form of the sample are very large, e.g., for Na_2_SO_4_, the SO_4_^2−^ ion content in the sample determined in the aqueous extract is ~2% while in the powdered sample it is ~7%. This gives a 300% difference in the results obtained depending on the determination method used. The highest consistency of results, regardless of the test method (EDS, XRF, UV or HPLC), was obtained for samples exposed to the MgCl_2_ salt. For this sample, the discrepancy between the results does not exceed 5%. 

When the NO_3_^−^ ion content is determined, the results obtained by testing aqueous extracts are identical. Due to the limitations of the XRF and EDS methods with regard to the detection of light elements, the content of nitrates was not determined in the solid phase. 

Discrepancies between results obtained by different methods (extract tests and powder tests) for the same sample depend on the type of salt. The variation of results for samples saturated with chloride salts is significantly smaller than for sulphate salts (Figure 11).

One reason for the lack of homogeneity of the results concerning the chloride and sulphate ion content determined in aqueous extracts and powder samples is the different solubility of each salt in water. Sodium, potassium and magnesium chloride salts have higher solubility in water than sulphate salts of the same metals. The solubility of magnesium, sodium and potassium chloride is 55 g, 36 g and 34 g/100 g H_2_O at 20 °C, respectively, whereas that of magnesium, sodium and potassium sulphate is 35 g, 20 g and 11 g/100 g H_2_O, respectively. On the other hand, the solubility of magnesium nitrate is the highest of the salts tested, at 70 g/100 g H_2_O. Of the chloride and sulphate salts tested, MgCl_2_ solubility is the highest. In the case of this salt, the results of the chloride content are very similar regardless of the test method used, and the difference between them is only 11% (Figure 11). The high solubility of magnesium chloride makes it easy to leach chloride salts from the ceramic matrix into the solution. This promotes the transition of all chlorides from the sample to the aqueous extract. In effect, the results for the chloride contents determined in the extract and in the solid phase (in the powder sample) are the same. However, in the case of low solubility, e.g., for sodium and potassium sulphate, the differences in results between samples (extract, solid sample) are very large. This means that some of the sulphates are not washed out and remain in the solid phase. They are not detected in the water extracts. Moreover, as the authors of the paper have shown [22], it is difficult to rinse the total amount of salt from the brick and is determined by such factors as fragmentation of the sample and the soaking time.

When analysing the results obtained for solubility of chloride and sulphate salts, note that the greater the solubility, the smaller the difference in results, and vice versa. However, this is not an unequivocal correlation, as is clearly seen in the case of the sample exposed to magnesium sulphate, the solubility of which is at level with that of chloride salts and yet the results vary depending on the test method. One reason for this is the permanent incorporation of some of the magnesium sulphate salts into the texture of the brick. In the studies presented in [11,22,23], it has been shown that sulphate ions introduced into ceramic brick in the form of a MgSO_4_ solution can react with calcium ions present in the matrix to produce sparingly soluble calcium sulphate (gypsum) and magnesium hydroxide. This is also possible with sodium and potassium sulphate [11,24]. The gypsum formed is a compound with a very low solubility of 0.26 g/100 g H_2_O. Therefore, it is extremely difficult to get into solution. Such a mechanism of incorporation of salt into the texture together with the formation of compounds of very low solubility is confirmed by the studies presented in this paper. Studies on chloride and sulphate content carried out on dried sediments (left after aqueous extracts) show that some of the salts, especially the sulphate salts, do not get into solution and remain in the solid phase. Tests of dried sediments (left after extracts) by XRF and EDS showed the presence of sulphur and chlorine in the amount which, when added to the sulphate and chloride content determined by UV and HPLC, gives the total content determined by XRF and EDS in powder samples (Figure 3). Taking into account the results obtained and their analysis, it can be concluded that each of the analytical methods used is correct. However, depending on the method of measurement (and in particular on the preparation tested), clearly different results can be obtained (for the same material) which should be interpreted differently.

In the analysis of salinity of building materials it is also important to determine the type of salt itself and not only the chloride and sulphate content. When analysing the curves obtained in the drying process (Figure 2), it was observed that depending on the metal cation (Na, K or Mg) in the chloride or sulphate salt, the drying process is different. For example, sodium sulphate causes a slight weight gain, whereas potassium sulphate causes a fourfold greater gain. Analysing the SEM image of the microstructure of the salt crystals formed, it can be concluded that one of the causes of the differentiation in the drying process may be the size and shape of the crystals formed. The microstructure image analysis shows that K_2_SO_4_ crystals are the largest among the salts studied and the Na_2_SO_4_ crystals are the smallest. Furthermore, an increase in the weight of bricks exposed to salt action may not only be the result of salt crystallisation but may also be the result of water retention in the material texture by hygroscopic salt. Prolonged moisture of walls has an adverse effect on the functional properties of buildings, hinders conservation works and also affects the mechanical parameters [25]. Moreover, depending on the cation, sulphate salts are characterised by different crystallisation pressure and different preferential crystallisation site [12,13,26,27]. Therefore, their impact on the durability of bricks will vary. The destructive action of salt is intensified by repeated recrystallisation of a given salt in the material texture as is the case with sodium sulphate [28] and magnesium sulphate [29]. Depending on the temperature at which the crystallisation process takes place, these salts have different degrees of hydration. The higher the degree of hydration, the greater the volume and higher crystallisation pressures [5,12,30]. The same is true for chloride salts, especially magnesium chloride. As demonstrated in [30], even small amounts of salt in the material may significantly affect the transport and hygroscopic properties of the material.

## 8. Conclusions

The investigation has shown that the content of a given salt determined in aqueous extracts by UV and HPLC is very similar. Similarly, when testing the salt content in powder samples by EDS and XRF, the results obtained are also very similar. However, the results in these two groups for a given salt may differ significantly, especially for sulphates. The discrepancies shown in the test results clearly show that the result may vary depending on the method of determination. For chloride salts, the discrepancies are significantly smaller than for sulphate salts. This clearly shows that the adopted test method is of great importance, and the interpretation of test results should always be related to the test method applied. When applying the UV and HPLC methods, note that the result does not necessarily reflect the total salt content. Where salts, especially sulphate salts, are found, it is appropriate and necessary to carry out additional tests, preferably under an electron microscope, together with an analysis of the elemental composition. This makes it possible to assess the total salt content together with the simultaneous identification of the type of salt which has been shown to be very important. This is because the aggressiveness of the salt can not only be associated with its dissolving capacity, but also with the crystallisation pressure, crystal size and the effect on the hygroscopic properties of the material. For this reason, it seems right and justified to recommend conducting salinity tests of walls in a comprehensive manner, enabling full identification of salt types and their content. It seems that the combination of spectrophotometric and scanning methods is the most optimal choice. The spectrophotometric method is a simple and relatively cheap method that does not require very expensive devices. The obtain results give information about the content of the most dangerous substances relative to the building materials, i.e., water soluble. A very important supplement to these tests is the observation in the scanning microscope that allows the identification of the type of phase whose solubility is very different. However, it is a method that requires expensive and specialised equipment as well as the knowledge of the person performing and interpreting the results.

## Figures and Tables

**Figure 1 materials-13-00559-f001:**
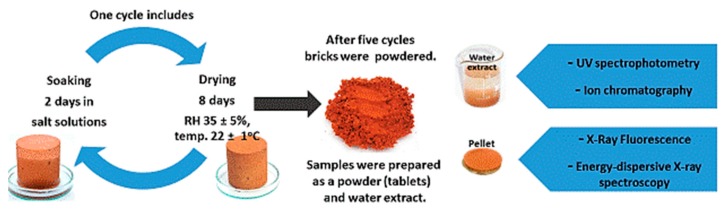
Scheme of soaking in salt solutions of MgCl_2_, KCl, NaCl, MgSO_4_, K_2_SO_4_, Na_2_SO_4_ and Mg(NO_3_)_2_ frame of researches. One cycle includes 2 d of soaking and 8 d of drying.

**Figure 2 materials-13-00559-f002:**
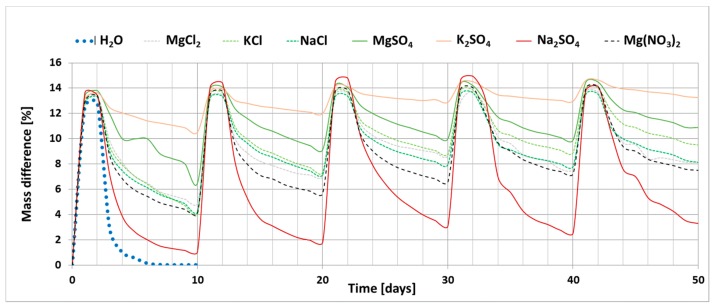
Change in the weight of samples saturated with salt solutions as a function of time (5 cycles).

**Figure 3 materials-13-00559-f003:**
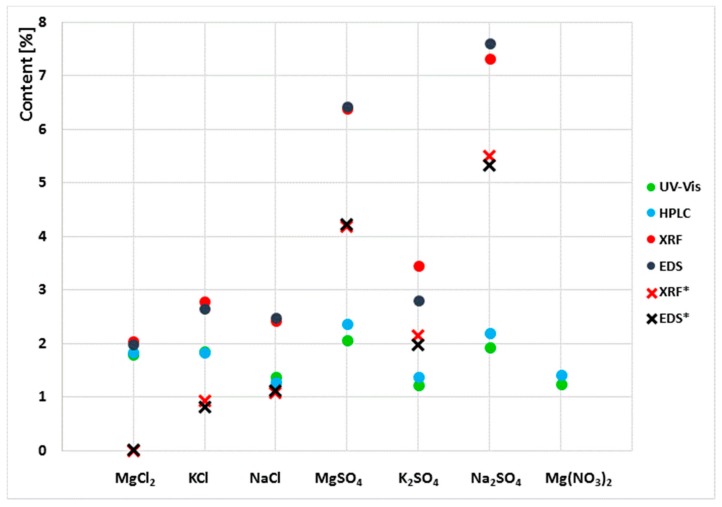
Results of chloride and sulphate ion content determined by liquid chromatography and chlorine and sulphur content determined by XRF and EDS. (*) denotes the results of the determination of the chlorine and sulphur content in the sediment after the leaching of sulphates and chlorides.

**Figure 4 materials-13-00559-f004:**
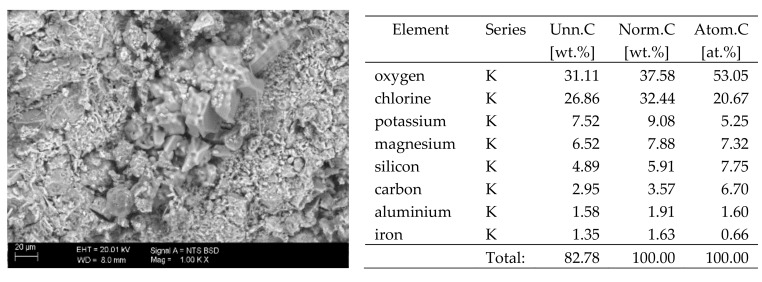
SEM image of the microstructure of MgCl_2_ crystals visible on the surface of the bricks tested; 1000× magnification. Next to it is a screenshot an EDS analysis for presence of crystals.

**Figure 5 materials-13-00559-f005:**
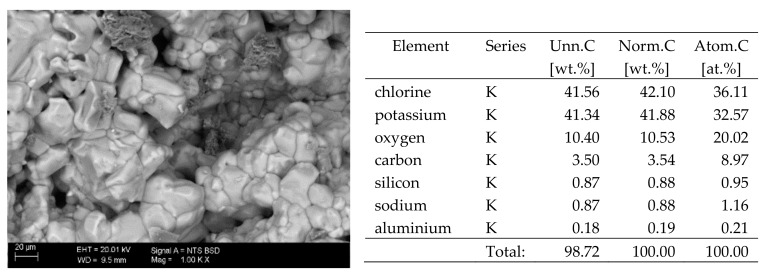
SEM image of the microstructure of KCl crystals visible on the surface of the bricks tested; 1000× magnification. Next to it is a screenshot an EDS analysis for presence of crystals.

**Figure 6 materials-13-00559-f006:**
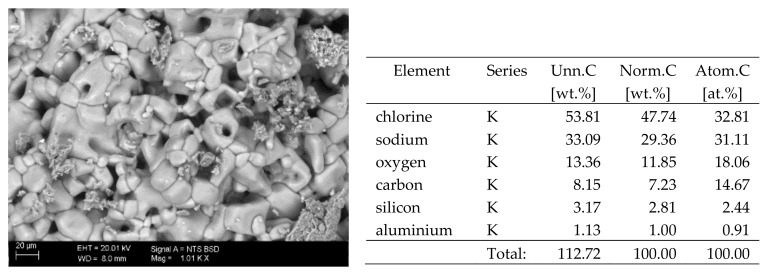
SEM image of the microstructure of NaCl crystals visible on the surface of the bricks tested; 1000× magnification. Next to it is a screenshot an EDS analysis for presence of crystals.

**Figure 7 materials-13-00559-f007:**
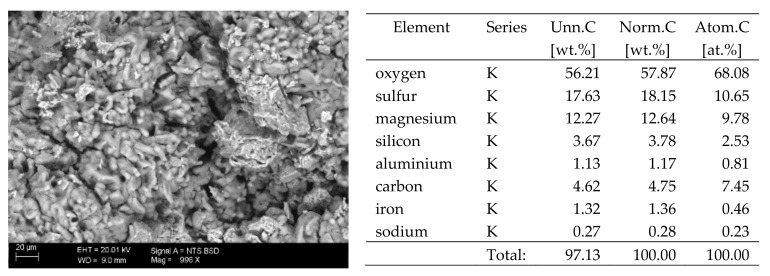
SEM image of the microstructure of MgSO_4_ crystals visible on the surface of the bricks tested; 1000× magnification. Next to it is a screenshot an EDS analysis for presence of crystals.

**Figure 8 materials-13-00559-f008:**
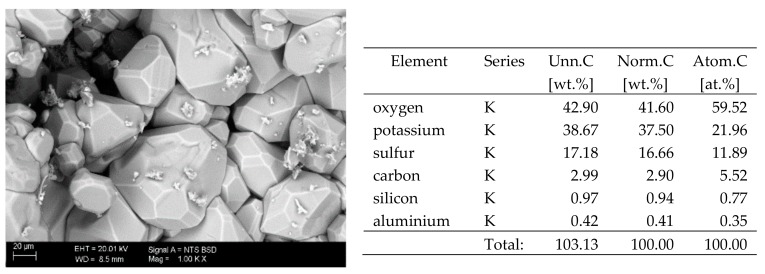
SEM image of the microstructure of K_2_SO_4_ crystals visible on the surface of the bricks tested; 1000× magnification. Next to it is a screenshot an EDS analysis for presence of crystals.

**Figure 9 materials-13-00559-f009:**
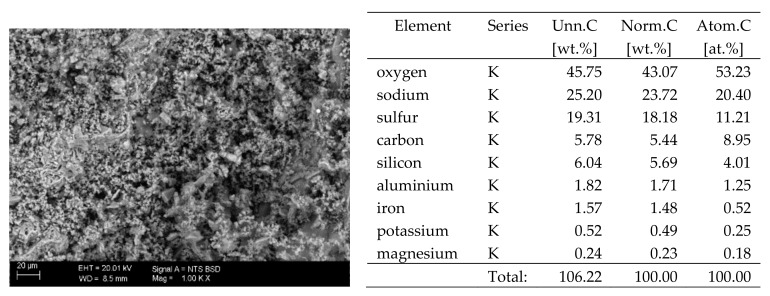
SEM image of the microstructure of Na_2_SO_4_ crystals visible on the surface of the bricks tested; 1000× magnification. Next to it is a screenshot an EDS analysis for presence of crystals.

**Figure 10 materials-13-00559-f010:**
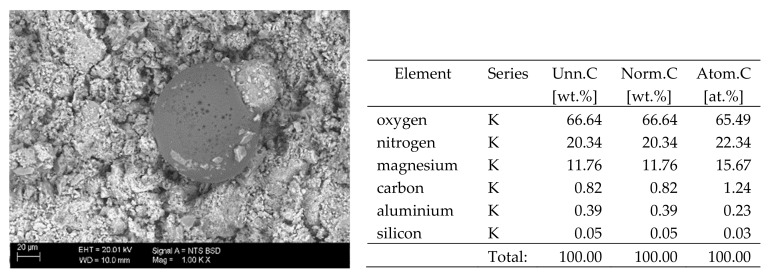
SEM image of the microstructure of Mg(NO_3_)_2_ crystals visible on the surface of the bricks tested; 1000× magnification. Next to it is a screenshot an EDS analysis for presence of crystals.

**Figure 11 materials-13-00559-f011:**
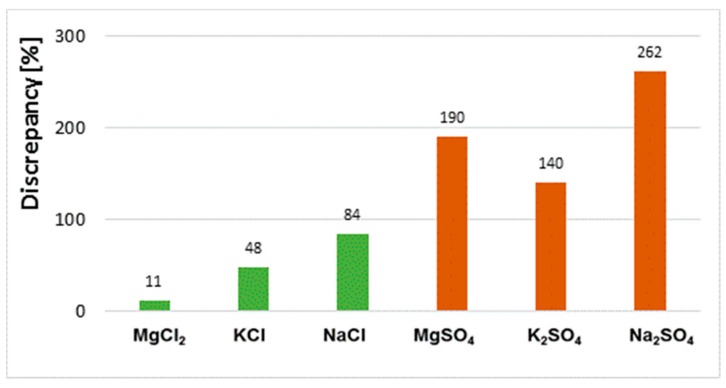
Discrepancy of results for chloride content in bricks exposed to MgCl_2_, KCl and NaCl salt solutions and for sulphate content in bricks exposed to MgSO_4_, K_2_SO_4_ and Na_2_SO_4_ salt solutions obtained in solid phase and aqueous extracts.

**Table 1 materials-13-00559-t001:** Results of tests for the content of selected ions and elements determined by different test methods.

Salts	UV	HPLC	XRF	EDS
	Content of ions Cl^−^ (%) mass		Content of element Cl (%) mass	
Reference material ^*(1)^	0.0029	0.0037	0.036	The margin of error methods
MgCl_2_	1.95	1.84	2.05	1.99
KCl	1.85	1.83	2.79	2.66
NaCl	1.38	1.29	2.43	2.49
	Content of ions SO_4_^2−^ (%) mass		Content of element S*^(2)^ (%) mass	
Reference material ^*(1)^	0.0106	0.0088	0.024	The margin of error methods
MgSO_4_	2.06	2.36	6.39	6.42
K_2_SO_4_	1.23	1.38	3.45	2.61
Na_2_SO_4_	1.93	2.19	7.32	7.61
	Content of ions NO_3_^−^ (%) mass		Content of element N (%) mass	
Reference material ^*(1)^	0.0029	0.0031	not tested *^(3)^	
Mg(NO_3_)_2_	1.37	1.41	not tested *^(3)^	

*(1) clean material, e.g., not subjected to saturation with salt solutions. *(2) sulphur content has been converted to SO_4_^2−^ ion. *(3) in the XRF and EDS methods the determination of light elements as nitrogen is erroneous.

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
