# Peer review of "Selection of Method of Chemical Analysis in Measuring the Salinity of Mineral Materials"

_materials, 2020, doi:10.3390/ma13030559_

Round 1
Reviewer 1 Report
The paper presents a comparison of the experimental determination of salinity using different known techniques in presence of different salts.
The paper is not innovative but it is usefull to understand the range of applicability of the different techniques in presence of different salts.
The conclusions should be more clear in assessing the applicability of the different techniques and the motivations for that.
The paper is well written and clear in its thread and aims.
Author Response
Dear Reviewer 1,
Thank you for your all remarks and comments, they were very useful. We could improve our text according your comments. If I was not clear (in our explanations), please let us know, and we will clarify further.
Minor corrections have been included in the text. In addition, we have completed the missing information on research methods. The conclusion have also been completed.
Reviewer 2 Report
General
The investigation deals with testing different analytical techniques for the determination of salinity in bricks used for buildings. It covers destructive methods with respect to the sample taken, like UV spectroscopy and ion chromatography (HPLC), for the determination of ionic species and non-destructive methods, like WDXRF and SEM-EDS, for multi-elemental analysis in the samples. The intention was to determine both the soluble and insoluble salt components. Brick samples subjected to accelerated salt weathering are used as test samples. The cyclic weathering processes were carried out by subjecting the bricks to 10% solutions of MgCl2, KCl, NaCl, MgSO4, K2SO4, Na2SO4 and Mg(NO3)2 under a laboratory room condition. The main purpose of the investigation is described as an attempt to assess the appropriateness of the analytical methods for the determination of the salinity in the brick samples and eventually select the most reliable one. In the case of the SEM-EDS analysis, in addition to the elemental composition determinations, morphological descriptions of the salt formations at the surface of the samples are given. The need to identify the most effective method (s) for the reliable determination of salinity in bricks and, generally building materials, is great. It is worthwhile to make comparisons between some of the methods selected, making the contribution valuable. Some of the analytical techniques that are relevant for the simultaneous identification and quantification of the mineralogy of salts, like X-ray diffraction (XRD), and for in-situ, non-destructive elemental analyses in a portable mode, such as energy dispersive X-ray florescence (EDXRF) and laser-induced breakdown spectroscopy (LIBS) are not compared to cite a few. The conclusion made doesn’t directly take into account the main objective stated from the outset, namely identification of a single reliable method, although the necessity to apply a combination of complementary techniques is implied. Other considerations such as the difficulty in sample preparation, representativeness of the measurements, time, cost and expertise needed, etc., could also be compared. Additional descriptions of the measurement parameters employed in many of the methods would make more informative how the analytical data were gathered, thereby facilitating the interpretations of data collected and replication of the measurements, among other advantages.
Specific comments and suggestions
Lines 24 and 25: This part addresses the main conclusion related to the objective of the work stated. Great if the appropriateness of the application of a combination of techniques emphasized as it is noted in the course of the investigation.
Line 31: Reference: formatting?
Line 33: …. literature of the subject matter, there is no ….
Line 38: Therefore, for tests of this type, the sample has to be …..
Line 48: … (EDS), together with ….
Line 51-53: When choosing the test methods, it is ….. accuracy if the determination, the measurement range, ….. representative sample), the time and cost of analysis.
Line 56: The mineral material? Is that to refer to aggressiveness of the crystallization of the salts (i.e., mineralogy of the salts)?
Line 58: …. as identification of the types of salts is essential.
Line 61: Good point. Why not then methods for the characterization of the mineralogy of the salt formations (like XRD) not incorporated in the method comparisons? It could also shed light on the possible insoluble salt formations, mentioned in the result part, due to the interactions of the saline solutions (used for the testing) with the diverse components in the bricks’ matrices.
Line 65-66: Was that to mean the other way around: ICP-OES and ICP-MS for total elemental analyses and ion chromatography for speciation (cations and anions)?
Line 78: … insoluble substances including phase identifications. But, which methods were for the phase identifications from the elemental determinations? Is that SEM-EDS as it combines the high-resolution imaging with elemental composition (the spot analyses)? Was mapping carried out?
Line 84: Then the spectrum of the sample …
Line 85 -86: … the substance in solution is based on the absorbance of the solution according to the Lambert- Beer law …..
Line 89: The concentration is the basic factor determining the amount of light absorbed at a given wavelength for a particular substance.
Line 91: …. consisted of fragmenting ….
Line 94 -95 (several minutes?) estimates roughly to be more specific.
Line 95: After that time, the mixture ….
Line 96: unit formatting
Line 97: Which ions were determined in this way and why? What wavelengths were selected for the determinations of the concentrations of the respective ions? The tray is mentioned, but what types of sample holders were used? Cuvette types and sizes used? How was the calibration performed? How many replicate measurements for assessing reproducibility? More description of the measurement conditions is essential for understanding how the data were eventually acquired.
Line 99: … individual components to identify and quantify them.
Line 119: ….. fluorescence spectrum … What size of the sample was used for the preparation of each pressed pellet? Was the ground sample sieved before pressing? What was the pressure used for preparation of the pellets?
Line 125: Unit formatting? Measurement and analysis conditions: kV, maximum current, dispersive crystals, filter, number of scans per sample, certified and reference materials used for calibration or standardless quantitative analysis aided by a software, etc. Were there replicate measurements where the average results reported?
Line 145: Multipoint scanning: selected area analyses? How was representativeness of the measurements ensured considering the heterogeneity of the test samples?
Line 151: Cut-out cylinders: How many were prepared in total for the different measurements?
Line 162: …. in salt solutions of …. K2SO4 and Mg(NO3)2 …
Since the weathering are not conducted in regulated weathering chambers, was the temperature and humidity of the laboratory room continuously monitored (logged data) during the course of the experiment? How often was the measurements taken?
Line 173: Complete water removal? The references used are the ‘dried’ samples at the prevailing laboratory room condition, not the dried ones, for example at 105 oC. The comparison is, thus, relative to the conditioned samples at the room temperature and humidity, not to the initial dry mass.
Line 186: In the ion chromatography examination, …..
In addition to the anions, were attempts made to identify and quantify the soluble cations by same method?
Line 193: What pressure was used for pressing? Pellet size prepared?
Line 195: It is understandable that the intention is to make comparisons with results from the anion determinations, however, the sulphur determined by these methods are the total elemental sulphur in both soluble and non-soluble forms. Moreover, the results displayed in table 1 are the chlorides and sulphates of magnesium, potassium and sodium. There is more conversion for the comparisons then.
Line 204: The reference material given in the table need some descriptions. What is it?
Line 205: Which light elements?
Line 209: Were these samples too prepared in same manner for the elemental analysis as the other brick samples subjected to the salt weathering?
Line 213: Quite reasonable since the XRF and EDS analyses determine the total elemental composition, whereas the extracts are associated only with the soluble ones in the specific species forms.
Line 220: I think the salt crystallizations at the surfaces of the samples examined to characterize their morphology and composition. What do the results look like from the EDS quantification carried out on the pressed pellets including imaging?
Line 225: Cross-sectional examinations, including the surface precipitations, could also shed light on other types of salts formed due to the integration of the saline solutions and the brick materials.
Line 227: Which area, from the BSE image displayed, was selected for the elemental quantification given? Whole area? The salt precipitated? Nice if some description of the area of analysis selected provided since the samples investigated are heterogeneous as shown in the images.
Line 231: Crystal clusters: Mixed salts implied from the simultaneous detection of sodium and potassium (NaCl + KCl)?
Line 232: The large carbon content: Implies carbonates of sodium? The additional results from anion ( also possibly cation) analyses in the ion chromatography could help to explain such observations. How about spot analyses on the different phases shown in the BSE image? Is the whole area displayed used for generating the elemental composition given? Were these measurements also conducted on non-coated dry samples under environmental/low vacuum condition? Carbon is present in all the results given (figures 4-10). Then, the measurement condition need to be specified to enable better interpretation. The silicates and aluminosilcates (Al, Si, O) appear to have originated from the brick.
Line 250: Interesting form: orthorhombic crystal structure?
Line 251-53: The composition is an indicator of the salt. The morphology is not that of anhydrous salt with well-defined structure. Anhydrous salt of magnesium nitrate has cubic crystal structure. The hexahydrate and mono-hydrate forms of the salt have other types of structures. Since it is highly hygroscopic in its nature, what is noted could possibly be supersaturated droplets, rather than crystals of magnesium nitrate.
Line 261: The observations are associated with the solubility of the salts\compounds of chloride and sulphate. The interaction of the sulphates with the brick materials is likely to form sparingly soluble salts as described later in this page and next page.
Line 269: PHPLC – a new abbreviation to be described in the manuscript?
Line 278: Measurement of EDS under low\high vacuum conditions: Could not allow nitrogen determination? Some results are given (figure 10). Even lighter elements than nitrogen are determined.
Line 302-3: They are not detected in the water extracts.
Line 363-5: The conclusion in line with aim set at the beginning: identification of one method? Not possible to find a single one considering the pros and cons of the analytical methods observed. It is also appropriate to consider the difficulty in sample preparation, representativeness of the measurement, time, cost and expertise needed to run the instruments. Other relevant methods could also be compared, like XRD for mineralogical investigation of the salts as well as cost effective and non-destructive methods that can be employed for in-situ measurements on site.
Author Response
Dear Reviewer 2,
Thank you for your all remarks and comments, they were very useful. We could improve our text according your comments. If I was not clear (in our explanations), please let us know, and we will clarify further.
Minor corrections have been included in the text. In addition, we have completed the missing information on research methods. The conclusion have also been completed. In the attachment we also send detailed information that answers your questions.
